# Innovative Methodology for Physical Modelling of Multi-Pass Wire Rod Rolling with the Use of a Variable Strain Scheme

**DOI:** 10.3390/ma16020578

**Published:** 2023-01-06

**Authors:** Konrad Błażej Laber

**Affiliations:** Department of Metallurgy and Metal Technology, Faculty of Production Engineering and Materials Technology, Czestochowa University of Technology, 19 Armii Krajowej Ave., 42-200 Czestochowa, Poland; konrad.laber@pcz.pl; Tel.: +48-34-325-07-97

**Keywords:** physical modelling, wire rod rolling, variable strain state, hot torsion test, metallographic tests, mechanical properties, cold upsetting steel

## Abstract

This paper presents the results of physical modelling of the process of multi-pass rolling of a wire rod with controlled, multi-stage cooling. The main goal of this study was to verify the possibility of using a torsion plastometer, which allows conducting tests on multi-sequence torsion, tensile, compression and in the so-called complex strain state to physically replicate the actual technological process. The advantage of the research methodology proposed in this paper in relation to work published so far, is its ability to replicate the entire deformation cycle while precisely preserving the temperature of the deformed material during individual stages of the reproduced technological process and its ability to quickly and accurately determine selected mechanical properties during a static tensile test. Changes in the most important parameters of the process (strain, strain rate, temperature, and yield stress) were analyzed for each variant. After physical modelling, the material was subjected to metallographic and hardness tests. Then, on the basis of mathematical models and using measurements of the average grain size, chemical composition, and hardness, the yield strength, ultimate tensile strength, and plasticity reserve were determined. The scope of the tests also included determining selected mechanical properties during a static tensile test. The obtained results were verified by comparing to results obtained under industrial conditions. The best variant was a variant consisting of physically replicating the rolling process in a bar rolling mill as multi-sequence non-free torsion; the rolling process in an NTM block (no twist mill) as non-free continuous torsion, with the total strain equal to the actual strain occurring at this stage of the technological process; and the rolling process in an RSM block (reducing and sizing mill) as tension, while maintaining the total strain value in this block. The differences between the most important mechanical parameters determined during a static tensile test of a wire rod under industrial conditions and the material after physical modelling were 1.5% for yield strength, approximately 6.1% for ultimate tensile strength, and approximately 4.1% for the relative reduction of the area in the fracture and plasticity reserve.

## 1. Introduction

The wire rod rolling process in modern rolling mills is characterized by high dynamics with the linear speeds of the rolled band reaching values of 120 m/s or more [1]. Such a high speed of the rolled band combined with short intervals between individual deformations results in the processes occurring in the material itself also becoming dynamic. The strain rate of the material often exceeds the value of 2000 s^−1^, which significantly hinders the physical modelling of such dynamic processes with the use of available research apparatus. Providing a required strain value, strain rate, and temperature during physical modelling that is similar to those occurring during the actual process ensures the high accuracy of obtained results [2]. These parameters have a direct impact on the shape and nature of the changes in the yield stress of the tested material, and thus the shaping of the microstructure and properties of the finished product [3,4]. In a situation where it is impossible to ensure any of the above-mentioned parameters are at the required level, one solution may be to adopt certain simplification assumptions. During the following stage, it should be verified whether such assumptions can be applied and how this affects the final result. Using certain simplifying assumptions depends on, among others, the specificity of the analyzed process and the tested material [2].

The most commonly used methods for the physical modelling of rolling processes are the compression test [5,6,7,8,9] and the torsion test [10,11,12,13,14]. These methods, despite the dynamic development of the laboratory equipment, have some limitations.

Most of the work concerning the physical modelling of the rolling processes published so far focuses on the assumption that the microstructure and properties of the final product are mainly determined by the few final passes, and the impact of the earlier stages of the technological process is less important [15]. As shown by the results of the research presented, inter alia, in the works [16,17,18,19,20,21], such a simplification is acceptable. Therefore, physical modelling studies are usually carried out only for the final stage of the analyzed technological process, often during compression tests by using the GLEEBLE metallurgical processes simulator [16,17,18,19,20,21]. The biggest advantage of this research methodology consists of the possibility of obtaining large values of strain rate, which significantly affects the level of yield stress. On the other hand, the biggest disadvantage is the limited value of the total strain value (about 1.2) and sometimes the limitations in the scope of precisely controlling the process of accelerated (controlled) cooling, which is important in the case of modelling processes which include the so-called multi-stage interoperational cooling. Moreover, after physical modelling, the material often cannot be used to determine mechanical properties directly during a tensile test, and determining the mechanical properties takes place indirectly based on measuring the grain size, hardness, and chemical composition [17,20,21], which is time-consuming and may include some error.

However, during physical modelling of the rolling process during a torsion test using a torsion plastometer, a limitation may be the low value of the strain rate. Apart from the standard free or non-free torsion tests, modern torsion plastometers enable testing in the course of multi-sequence torsion (including alternating), tensile, and compression tests, and in the so-called complex strain state test (simultaneous torsion with compression or simultaneous torsion with tensile tests). Furthermore, they enable precise temperature control during individual stages of the reproduced technological process, and a sole torsion test allows for much greater deformation values than a compression test.

Most of the works published so far on the physical modelling of real technological processes in torsion tests refer to the rolling of sheet metal bars [12,22,23,24,25], stepped shafts [26], or service pipes [10]. Therefore, it is reasonable to conduct research on solving problems related to the physical modelling of wire rod rolling processes in modern rolling mills, which are characterized by high linear velocities of the rolled strand with the use of modern torsion plastometers, enabling testing of the variable strain state.

This paper presents the results of physical modelling of the process of multi-pass wire rod rolling with controlled, multi-stage cooling. The main objective of the study was to verify the possibility of using a torsion plastometer, enabling testing with the use of a variable strain scheme to physically reproduce the technological process of rolling a wire rod.

The advantage of the research methodology proposed in the paper in reference to the works published so far is the possibility of reproducing the entire strain cycle (with a total strain value of 14.32), precisely preserving the temperature value of the deformed material during individual stages of the reproduced technological process, and the possibility of quickly and accurately determining selected mechanical properties during a static tensile test directly from the material after physical modelling.

The tests presented in the paper were carried out on several variants with the use of the STD 812 torsion plastometer [27], which, in addition to standard free or non-free torsion tests, allows conducting tests during multi-sequence torsion (also alternating), tensile, and compression tests, and also in the so-called complex strain state test (simultaneous torsion with compression or simultaneous torsion with tensile test).

For each variant, changes in the most important process parameters (strain, strain rate, temperature, and yield stress) were analyzed in detail. After physical modelling, each time the material was subjected to metallographic tests at characteristic points in order to assess the microstructure of the tested steel and the average ferrite grain size. In addition, Vickers hardness tests were also carried out. Then, based on the available mathematical models and using the measurements of the average ferrite grain size, chemical composition, and hardness [28,29], the yield strength, ultimate tensile strength, and plasticity reserve were determined. The scope of research carried out as part of the work also included determining selected mechanical properties directly in the static tensile test. The obtained results were verified under industrial conditions.

Basing on the obtained research results, it has been determined that the best variant is a variant consisting of physically reproducing the rolling process in a bar rolling mill as multi-sequence non-free torsion; the rolling process in an NTM block (No-Twist Mill) as non-free continuous torsion, with the total strain equal to the actual strain occurring at this stage of the technological process; and the physical modelling of the rolling process in an RSM block (reducing and sizing mill) as a tension, while maintaining the total strain value in this block. The differences between the most important mechanical parameters determined during a wire rod static tensile test under industrial conditions and during tensile tests of material after physical modelling were 1.5% for the yield strength, approximately 6.1% for the ultimate tensile strength, and approximately 4.1% for the relative reduction of area at fracture and the plasticity reserve.

According to the authors, such high accuracy between the research results obtained under industrial and experimental conditions results mainly from inducing a high level of yield stress during the final stage of physical modelling of the analyzed process (RSM block), by changing the deformation state from torsion to tension. The level of yield stress obtained in such a manner is close to the value of the yield stress occurring in the actual technological process, which is crucial for activating the microstructure rebuilding processes.

## 2. Materials and Methods

### 2.1. Materials

The research presented in the paper was carried out for low-carbon cold upsetting steel of the 20MnB4 grade, with a chemical composition in accordance with PN-EN 10263-4:2004 (Table 1) [17,21,30].

### 2.2. Methods

Physical modelling tests were carried out for the entire production cycle of rolling a wire rod with a final diameter of 5.5 mm, for an exemplary combined-type rolling mill (a combination of a bar rolling mill and a wire rod rolling mill) (Figure 1). The rolling process in a continuous rolling mill took place over 17 passes, while rolling in a wire rod rolling mill took place in 2 blocks: a No-Twist Mill (NTM) 10-rolling block, and a reducing and sizing mill (RSM) 4-rolling block.

However, during tests it is necessary to take into consideration the temperature increase caused by the deformation of the material in the RSM block with a high strain rate, which for the tested steel grade was about 50 °C. Therefore, during the tests, the temperature of the rolled band in the RSM block was 850 °C. After the rolling process, the material was cooled in two stages. During the first stage of the controlled cooling process, the 20MnB4 steel was cooled from the rolling end temperature to 500 °C at a cooling rate of approx. 10 °C/s, while during the second stage of cooling, the tested material was cooled to 200 °C at a cooling rate of 1 °C/s.

Several different tests were carried out in this paper with the use of the STD 812 torsion plastometer [27], which, in addition to standard free or non-free torsion tests, allows conducting tests during multi-sequence torsion (also alternating), tensile, and compression tests, and in the so-called complex strain state test (simultaneous torsion with compression or simultaneous torsion with tensile test). For the tests, circular samples with dimensions of diameter, d, of 9 mm and length, l, of 2 mm were used (Figure 2). An S-type thermocouple (PtRh10-Pt) welded to the side surface of the sample was used to register and control changes in temperature. The general view of the test chamber and the main parameters of the device are presented in Figure 3.

In order to determine the actual strain value, the relation (1) was used, the actual strain rate was determined based on the relation (2), while the yield stress was calculated according to the Formula (3) [36,37]:(1)$$\varepsilon =\frac{2\cdot \pi \cdot r\cdot N}{\sqrt{3}\cdot L}$$
(2)$$\overset{•}{\varepsilon }=\frac{2\cdot \pi \cdot r\cdot \overset{•}{N}}{\sqrt{3}\cdot 60\cdot L}$$
(3)$$\sigma_{p}=\frac{\sqrt{3}\cdot 3M}{2\pi r^{3}}$$
where *r* is the sample radius, *L* is the sample length, *N* is the number of sample twists (revolutions), $\overset{•}{N}$ is the torsion speed (rpm), and *M* is the torque.

Research concerning the physical modelling of the wire rod rolling process was carried out in three variants, differing mainly in the method of applying deformation.

**Variant 1 (V1):** Physical modelling of the analyzed process in this variant was carried out during non-free torsion tests, as a cycle of 31 individual deformations at a certain temperature with specified intervals between successive deformations, and taking into account multi-stage controlled cooling during the individual stages of the technological process:-*17 individual deformations as non-free torsion, representing the rolling process in a continuous bar rolling mill;*-*10 individual deformations as non-free torsion, representing the rolling process in an NTM block of a wire rod rolling mill;*-*4 individual deformations as non-free torsion, representing the rolling process in an RSM block of a wire rod rolling mill.*

**Variant 2 (V2):** The physical modelling of the analyzed process in this variant was carried out in non-free torsion tests, as a cycle of 19 deformations at a certain temperature with specified intervals between following deformation cycles, and taking into account multi-stage controlled cooling at individual stages of the technological process:-*17 individual deformations as non-free torsion, representing the rolling process in a continuous bar rolling mill;*-*1 deformation reproducing the rolling process at an NTM block of a wire rod rolling mill, as non-free torsion, with a total strain value equal to the actual strain value occurring at this stage of the technological process;*-*1 deformation reproducing the rolling process at an RSM block of a wire rod rolling mill, as non-free torsion, with a total strain value equal to the actual strain value occurring in this block.*

**Variant 3 (V3):** The physical modelling of the analyzed process in this variant was carried out during non-free torsion and tensile tests, as a cycle of 19 deformations at a specific temperature with specified intervals between successive deformation cycles, and taking into account multi-stage controlled cooling during individual stages of the technological process:-*17 individual deformations as non-free torsion, representing the rolling process in a continuous bar rolling mill,*-*1 deformation reproducing the rolling process in an NTM block of a wire rod rolling mill, as non-free torsion, with a total strain value equal to the actual strain value occurring at this stage of the technological process and,*-*1 deformation reproducing the rolling process in an RSM block of a wire rod rolling mill, as tension, while maintaining the total strain value in this block.*

The general diagram of thermo-mechanical treatment, reproducing the entire rolling process of a 20 MnB4 steel wire rod with a diameter of 5.5 mm, is shown in Figure 4.

Before the deformation process, the material was heated to a temperature of 1165 °C, corresponding to the temperature in the equalizing zone of a heating furnace (under industrial conditions). Then, in order to unify the temperature distribution in the entire sample working zone, the 20MnB4 steel was heated for 300 s. The following stage consisted of cooling for 30 s to a temperature of 1086 °C, replicating the cooling of the band during its transport from the furnace to the first rolling pass stand. Then, the tested samples were deformed over 17 cycles with strain parameters in accordance with Table 2, replicating the rolling process in a continuous rolling mill. The following stage consisted of accelerated cooling to 851 °C, corresponding to the band temperature before the NTM block of a rolling mill. During the following stage of physical modelling, the rolling process in an NTM block was reproduced (Table 3). Then, the process of accelerated cooling between NTM and RSM blocks was modelled, with cooling to a temperature of 845 °C, corresponding to the band temperature before the RSM block. The next stage of physical modelling consisted of reproducing the rolling process in an RSM block (Table 3). During the final stage, the tested steel grade was cooled in a controlled manner to 500 °C with an average cooling rate of about 10 °C/s, and then to 200 °C with a cooling rate of 1 °C/s.

After physical modelling, changes in the most important parameters of the process (strain, strain rate, temperature, and yield stress) were analyzed in detail.

During the second stage of work, samples were cut out from the material for metallographic tests after physical modelling using an EDM 32 wire electro-driller, and then metallographic microsections (nitrile etching) were prepared (Figure 5).

The locations of characteristic points where metallographic analyses and hardness measurements were carried out are presented in Figure 6 (r = radius).

An assessment of the microstructure of the tested steel was carried out using light microscopy (Nikon Eclipse MA-200 microscope with NIS-Elements software) [38]. The average ferrite grain size was determined using the perpendicular secant method [39]. The hardness tests were carried out using the Vickers method with the use of a Future-Tech FM-700 microhardness meter (load 1000 gf, time 5 s).

In the following stage, on the basis of available mathematical models and using measurements of average ferrite grain sizes, chemical composition, and hardness [28,29], the yield strength (*YS*), ultimate tensile strength (*UTS*), and plasticity reserve were determined.
(4)$$YS_{}=\frac{HV}{0.378}-123,$$
(5)$$UTS=\frac{HV}{0.352}+70,$$
(6)$$YS=62.6+26(\%Mn)+60(\%Si)+759(\%P)+213(\%Cu)+3286(\%N)+\frac{19.7}{\sqrt{\frac{D_{\alpha }}{1000}}},$$
(7)$$UTS=165+54(\%Mn)+100(\%Si)+652(\%P)+473(\%Ni)+635(\%C)+2173(\%N)+\frac{11}{\sqrt{\frac{D_{\alpha }}{1000}}},$$
where *HV* is the Vickers hardness; %Mn, %Si, %P, %Cu, %N, %Ni, and %C are the content in mass percent of manganese, silicon, phosphorus, copper, nitrogen, nickel, and carbon, respectively, in the steel; and *D_α_* is the ferrite grain size in μm.

The scope of research carried out in terms of the work also included determining selected mechanical properties directly during the static tensile test using a Zwick Z/100 strength machine [38].

During the final stage of the work, the obtained research results were verified under industrial conditions.

## 3. Results and Discussion

### 3.1. Analysis of the Main Parameters of the Deformation Process

The general course of temperature and plasticizing yield stress during the physical modelling of the rolling process of 20MnB4 steel round bars in a medium continuous rolling mill is shown in Figure 7. This stage of physical modelling was the same for all three variants analyzed in the work.

The most important parameters of the deformation process during the physical modelling of rolling of round bars made of 20MnB4 steel in a medium continuous rolling mill, which have been tested, determined, and verified in previous works [17,21], are presented in Table 2. The “/” symbol is followed by parameters that can be achieved by the STD 812 torsion plastometer, taking into account its technical parameters mainly inertia.

When analyzing the data presented in Figure 7 and in Table 2, a decrease in the temperature of the band during the initial stage of the deformation process (deformations 1–7) and a simultaneous increase in the value of the yield stress to 84.9 MPa (for pass No. 7) were determined. This resulted from the long intervals between the deformations and the low strain rate value. Analyzing the remaining deformations replicating the rolling process in a continuous rolling mill (No. 8–17), it is possible to notice a gradual increase in the temperature of the deformed steel, caused, among others, by an increase in the rolling speed, and thus shorter intervals between subsequent deformations. There is a general downward trend in the yield stress to a value of 78.9 MPa (pass No. 17). Comparing the values of the nominal (set) temperature and that obtained during the physical modelling of the rolling process of 20MnB4 steel in a bar rolling mill, a high accuracy for this analyzed parameter was found. The biggest error was between the set temperature value and the obtained value in pass No. 16, at 0.6%.

Analyzing the data in Table 2, it can be noted that starting from pass No. 6, it was not possible to maintain the required strain rate value. This was due to the inertia of the device. The time needed to reach the required strain rate value was about 0.25 s. Using higher strain rate values than 1.15 s^−1^, with values of individual deformations of 0.5–0.6, resulted in exceeding the strain value. According to published data, among others, in the paper [2], despite a great difference between the desired value of strain rate and the value achievable by the torsion plastometer, the differences in the values of yield stress did not exceed 6.3%, which did not have a significant impact on the microstructure and mechanical properties of the tested material after this stage of the rolling process [2].

The minimum time between deformations achievable by a torsion plastometer was 0.9 s. After the last deformation in the rod rolling mill (No. 17), in accordance with the industrial conditions, the material was cooled to a temperature of about 851 °C, required before the first deformation in an NTM block of a wire rod rolling mill (Table 3). Accelerated cooling was carried out using argon supplied by special nozzles to the central part of the sample area (Figure 3a, in the case of slow cooling rate, the device simultaneously cools down and heats the tested material to ensure the required temperature value). The accelerated cooling time was increased from about 8.5 s to about 55 s (cooling over 40 s and holding 15 s (Table 2)). Increasing this time and holding at a temperature of about 851 °C was necessary due to the inductive method of heating the samples and the shape of the inductive exciter itself (Figure 3a). Using a cooling time in accordance with industrial conditions (approx. 8.5 s) after reaching the required temperature (851 °C) and shutting down the cooling system resulted in a rapid increase in temperature in the middle part of the samples (Figure 8a). This increase was caused by heat conduction towards the central part of the samples from areas with a higher temperature. This resulted in the inability to achieve the required temperature value during the initial stage of the deformation process in an NTM block. The difference between the nominal temperature value and the desired value was 4.7% (Figure 8a).

The general course of temperature and yield stress during the physical modelling of a 20MnB4 steel wire rod rolling process in NTM and RSM blocks of a wire rod rolling mill for the V1 test variant is shown in Figure 9.

The most important parameters of the deformation process during the physical modelling of rolling a 20MnB4 steel wire rod with a diameter of 5.5 mm in NTM and RSM blocks of a wire rod rolling mill according to the V1 test variant are presented in Table 3. As before, the “/” symbol is followed by parameters that can be achieved by the STD 812 torsion plastometer, taking into account its technical parameters.

Analyzing the temperature changes during the deformation reproducing the rolling process in an NTM block (No. 18–27), it is possible to observe a continuous increase in temperature caused mainly by short intervals between successive deformations. Comparing the nominal (set) and obtained temperature values during the physical modelling of the 20MnB4 steel rolling process in an NTM block, a high accuracy of this analyzed parameter was found. The biggest error between the set temperature value and the obtained value occurred for passes No. 18 and 22, at 0.2%.

The course of yield stress during the first five deformation sequences at this stage of the physical modelling process is somewhat unusual. As the temperature increases, the yield stress decreases and then slightly increases. This may result from an uneven temperature distribution after the process of accelerated cooling between the continuous rolling mill and the NTM block. During subsequent cycles of deformations simulating the rolling process in an NTM block, the course of yield stress is typical, i.e., as the temperature of the deformed material increases, the yield stress value of the deformed steel decreases.

The minimum possible time between deformations achieved by the torsion plastometer was 2 s. After the final deformation in an NTM block (No. 27), in accordance with industrial conditions, accelerated material cooling to a temperature of about 845 °C was required before the first deformation in an RSM block of a wire rod rolling mill was applied (Table 3). The accelerated cooling time, similar to the accelerated cooling after a continuous rolling mill (pass No. 17), was increased from approx. 0.82 s to approx. 45 s (cooling over 30 s and holding 15 s to ensure the required temperature values in the RSM block).

Analyzing the temperature changes during the deformation reproducing the rolling process in an RSM block (No. 28–31), it is possible to observe (similar to the case of deformation in the NTM block) a continuous increase in temperature caused mainly by short intervals between successive deformations. By comparing the values of the nominal (set) temperature and that obtained during the physical modelling of the 20MnB4 steel rolling process in an RSM block, a high accuracy of this analyzed parameter was found. The largest error between the set temperature value and the obtained value occurred for pass No. 28 and was 0.4%. Despite an increase in temperature during the first two deformation sequences (No. 28 and 29), the yield stress increased (Table 3), which may result from an uneven distribution of temperature after the process of accelerated cooling between the NTM and RSM blocks. A decrease in the yield stress value was observed for the two final deformation sequences (No. 30 and 31). This resulted mainly from a small strain value of about 0.1. The minimum time between deformations achievable by the torsion plastometer during this stage of physical modelling was 0.9 s.

The general course of temperature and yield stress during the physical modelling of 20MnB4 steel wire rod rolling process in NTM and RSM blocks of a wire rod rolling mill for the V2 test variant is shown in Figure 10. The most important parameters of the deformation process during the physical modelling of rolling a 20MnB4 steel wire rod with a diameter of 5.5 mm in NTM and RSM blocks of a wire rod rolling mill according to the V2 test variant are presented in Table 4. As before, the “/” symbol is followed by parameters that can be achieved by the STD 812 torsion plastometer, taking into account its technical parameters. In this variant, the sequences of individual passes in NTM and RSM blocks of a wire rod rolling mill were replaced with individual deformations as non-free torsion, with a total strain value equal to the actual strain value occurring in these blocks. Such an assumption was tested during research presented, for example, in works [17,40]. Based on the obtained results, it was found that replacing the sequence of the four final deformations with one did not cause a large error in the distribution of yield stress and did not incur large errors in terms of assessing the structural construction of the samples.

Replacing the sequence of individual deformations in NTM and RSM blocks of a wire rod rolling mill with single deformations with strain values equal to the actual strain value occurring in these blocks allowed for the use of a higher value of the strain rate compared to the test variant V1. In the case of the NTM block, the strain rate was 40 s^−1^, whereas in the case of the RSM block it was 10 s^−1^. In turn, this had a positive impact in that it increased the value of the yield stress. In relation to the V1 test variant, the yield stress in the NTM block increased in value to about 223 MPa, while in the case of the RSM block, the yield stress value increased to approximately 145 MPa. During the physical modelling of the rolling process in NTM and RSM blocks according to the V2 test variant, the increase in temperature in these blocks was also programmed. Taking into account the inductive method of heating the samples and the shape of the inductive exciter itself (Figure 3a), in order to increase the accuracy between the assumed and the obtained temperature, in this variant the speed of accelerated cooling was increased between the continuous rolling mill and the NTM block as well as between the NTM and RSM blocks, and the resistance immediately after accelerated cooling was abandoned. This facilitated an increase in temperature as a result of the rapid thermal conductivity from areas with higher temperatures towards the central part of the samples. The error between the temperature assumed after deformation in the NTM block and the temperature obtained during physical modelling was 10%. In the case of the RSM block, the required temperature increase was achieved.

The general course of temperature and yield stress during the physical modelling of a 20MnB4 steel wire rod rolling process in NTM and RSM blocks of a wire rod rolling mill for the test variant V3 is shown in Figure 11.

The most important parameters of the deformation process during the physical modelling of rolling a 20MnB4 steel wire rod with a diameter of 5.5 mm in NTM and RSM blocks of a wire rod rolling mill according to the V3 test variant are presented in Table 5. The “/” symbol is followed by the parameters that can be achieved by the STD 812 torsion plastometer, taking into account its technical parameters.

In this variant, the sequences of individual passes in NTM and RSM blocks of a wire rod rolling mill were replaced by single deformations: one deformation reproducing the rolling process in an NTM block of a wire rod rolling mill as non-free torsion, with the total strain value equal to the actual strain value occurring during this stage of the technological process, and one deformation reproducing the rolling process in an RSM block of a wire rod rolling mill as tension, while maintaining the total strain value in this block.

Replacing the sequence of individual deformations in NTM and RSM blocks of a wire rod rolling mill with single deformations with strain values equal to the actual strain value occurring in these blocks (similar to the case of variant V2) allowed using a higher value of the strain rate in relation to the V1 test variant. In the case of the NTM block, the strain rate was 40 s^−1^, whereas in the case of the RSM block, it was 10 s^−1^. In turn, this had a positive impact in that there was an increase in the value of the yield stress. The yield stress in the NTM block reached approximately 191 MPa, while in the case of the RSM block, the yield stress increased to approximately 500 MPa, mainly as a result of changing the strain state. Similar to the case of the V2 variant, a temperature increase in the NTM and RSM blocks was also programmed in this variant. In order to increase the accuracy between the assumed and the obtained temperature, in this variant (similar to variant V2) the speed of accelerated cooling was increased between the continuous rolling mill and the NTM block as well as between the NTM and RSM blocks, and the resistance immediately after accelerated cooling was abandoned. This facilitated an increase in temperature as a result of the rapid thermal conductivity from areas with higher temperatures towards the central part of the samples. The error between the temperature assumed after deformation in the NTM block and the temperature obtained during physical modelling was 7.4%. In the case of the RSM block, the error between the temperature assumed after deformation and the temperature obtained during physical modelling was less than 2%.

It was found that the obtained level of yield stress, as a result of replacing the sequence of individual deformations in NTM and RSM blocks with one deformation and changing the strain state in the RSM block, is similar to the value of yield stress occurring in the actual technological process [17], which is crucial for the process of microstructure reconstruction.

The general course of changes in yield stress for all analyzed variants is shown in Figure 12.

### 3.2. Impact of the Used Deformation Process Conditions on the Microstructure of the Tested Material

Examples of the 20MnB4 steel microstructure after the physical modelling process according to the V1–V3 variants are shown in Figure 13, Figure 14 and Figure 15. The results of measuring the ferrite grain size and the hardness of the tested steel grade are presented in Table 6 and Figure 16.

Based on the analysis of the test results of 20MnB4 steel after physical modelling of the rolling process of a wire rod according to the test variant V1, it was found that the D_α_ ferrite grain size was between 8.14 and 5.48 μm. The average D_α_ ferrite grain size was 6.43 μm (Table 6, Figure 16). The hardness of the tested material after physical modelling according to this variant ranged from 179.75 to 216.33 HV (average value 205.19 HV).

The D_α_ ferrite grain size for the material after physical modelling in accordance with variant V2 was in the range of 7.31 to 5.7 μm. The average D_α_ ferrite grain size was only slightly larger (comparably) to the value achieved as a result of modelling in accordance with the V1 variant and was at 6.62 μm (Table 6 and Figure 16). This was due to an only slightly larger value of the yield stress during the physical modelling of the deformation process in the RSM block (despite an increase in the yield stress during the physical modelling of the deformation process in the NTM block in this variant in relation to the V1 variant, Figure 12). The hardness of the tested steel after physical modelling according to the technological variant V2 ranged from 182.37 to 213.43 HV (average value 203.26 HV).

Analyzing the test results for variant V3, it was determined that the D_α_ ferrite grain size of 20MnB4 steel was in the range of 8.31 to 6.16 μm. The average D_α_ ferrite grain size was 7.20 μm (Table 6, Figure 16).

The increase in the average D_α_ ferrite size resulted mainly from the much higher value of the yield stress during the physical modelling of the deformation process in the RSM block, mainly due to the change in strain state from non-free torsion to tension (and the increase in the strain rate). Moreover, the value of yield stress during the physical modelling of the deformation process in the NTM block in this variant in relation to the V1 variant was also higher (Figure 12). After physical modelling according to the technological variant V3, the hardness of 20MnB4 steel ranged from 158.30 to 221.80 HV, while the average hardness value was 202.30 HV.

Based on an analysis of the D_α_ ferrite grain size distribution in a cross-section (along the radius) (Figure 16), it was determined that the largest D_α_ ferrite grains occurred along the axis of the tested samples, while the smallest D_α_ ferrite grains occurred in subsurface areas. This is due to the characteristics of the torsion test itself, in which the smallest strain value occurs in the axis of the material subject to torsion, while the largest strain value occurs in the subsurface areas. Based on the data presented in Table 6 and Figure 16, it was determined that there was a simultaneous increase in the hardness of the tested steel along with a decrease in D_α_ ferrite grain size.

Analyzing the distribution of D_α_ ferrite grain size in a cross-section of the samples (along the radius), it is possible to observe a relatively high homogeneity in size. This may result from the high value of the total strain value (14.32) and the small length of the sample’s measured part (2 mm, total torsion angle of about 635°), whereas by analyzing the grain size at individual measurement points (Figure 13, Figure 14 and Figure 15) (in accordance with Figure 6), it is possible to notice a relatively large heterogeneity and the acicular shape of grains in certain areas. This may be due to the relatively high cooling rate after the deformation process (10 °C/s). According to the research results presented in paper [17], the applied cooling speed is the cooling speed limit for 20MnB4 steel, and if this value is exceeded, bainitic structures begin to form in the material.

Comparing the ferrite grain size of a wire rod produced in industrial conditions with the values obtained as a result of our physical modelling, it was found that the best results were obtained using physical modelling according to variant V3. The average size of the ferrite grain in the cross-section of the wire rod produced under industrial conditions was 7.52 μm. However, the average size of the ferrite grain in the longitudinal cross-section of the finished product was equal to 8.11 μm. The error between the ferrite grain sizes measured in a wire rod cross-section and obtained as a result of physical modelling according to the technological variant V3 was 4.3%, whereas the error between the ferrite grain sizes measured on the longitudinal cross-section of a wire rod and obtained as a result of physical modelling for this technological variant was 11.2%. On this basis, it can be concluded that the average ferrite grain size in a wire rod obtained under industrial conditions is similar to the grain size obtained in samples after physical modelling of the rolling process according to variant V3.

### 3.3. The Impact of the Applied Conditions of the Deformation Process on the Selected Mechanical Properties of the Tested Steel

Table 7 presents the results of research concerning selected mechanical properties of the material after physical modelling of the rolling process of a 20MnB4 steel wire rod with a diameter of 5.5 mm calculated using Formulas (4)–(7). These properties were determined from average hardness values and ferrite grain size. Moreover, this table presents the results of selected mechanical properties of the material after physical modelling, determined in the course of a static tensile test. This table also includes the results of selected mechanical properties of 20MnB4 steel obtained after rolling under industrial conditions. Additionally, this table also includes the results of tests obtained in previous studies [17,21], which carried out modelling only in the final stage of the analyzed rolling process (RSM block) in compression tests, using the GLEEBLE 3800 metallurgical process simulator.

A general view of the 20MnB4 steel samples after physical modelling of the process of rolling a wire rod during testing mechanical properties in a static tensile test is presented in Figure 17a. Examples of samples before and after the tests are shown in Figure 17b, while examples of tensile curves are shown in Figure 17c.

Based on an analysis of the results of mechanical properties tests (Table 7), it was determined that applying the modifications to the physical modelling of the wire rod rolling process resulted in a decrease in the yield strength and ultimate tensile strength of the tested steel, regardless of the applied analytical Formulas (4)–(7). This was also confirmed by the results of static tensile tests. No changes in values were observed with regard to the plasticity reserve. Due to the small length of the test area of the used samples (2 mm) and the inability to install an accurate extensometer, the relative elongation was not tested during the static tensile test. Analyzing the obtained test results concerning mechanical properties, it was found that the best research variant allowing to obtain a high correspondence between the mechanical properties determined in industrial research and for the material after physical modelling is the research variant V3. The differences between the most important mechanical parameters determined during a static tensile test of a wire rod under industrial conditions and material after physical modelling were 1.5% for the yield strength, approximately 6.1% for the ultimate tensile strength, and approximately 4.1% for the relative reduction of area at fracture and plasticity reserve. The high consistency between the mechanical properties determined in industrial research and for the material after physical modelling according to variant V3 was also confirmed by the results of calculations of selected mechanical properties using Equations (4)–(7).

When comparing the obtained results of mechanical properties tests (static tensile test for variant V3) and those presented in earlier works [17,21] with the results of industrial research (Table 7), it was found that the lowest accuracy was provided by Equations (4) and (5). The differences between the analyzed mechanical properties determined using these formulas and the values obtained in industrial conditions were 2.7% for the yield strength, approximately 13.5% for the ultimate tensile strength, and approximately 14.9% for the plasticity reserve. This may be due to the fact that these relations have been developed only for steel with a specific chemical composition. Moreover, these dependencies are based only on hardness measurements, which can be burdened with a certain error. A much higher accuracy was obtained using Equations (6) and (7) and a static tensile test. The differences between the analyzed mechanical properties determined using formulas (6) and (7) and the values obtained in industrial conditions were 3.6% for the yield strength, approximately 1.3% for the ultimate tensile strength, and approximately 2.7% for the plasticity reserve. The differences between the most important mechanical parameters determined during a static tensile test of a wire rod under industrial conditions and material after physical modelling were 1.5% for the yield strength, approximately 6.1% for the ultimate tensile strength, and approximately 4.1% for the relative reduction of area at fracture and plasticity reserve. Results obtained using these two test methods are comparable. The greater accuracy of results of mechanical properties tests obtained using Equations (6) and (7) may result from the fact that these relations take into account the chemical composition of a particular steel grade and ferrite grain size. An advantage of the research methodology proposed in this paper in relation to the works published so far is that it can quickly and accurately determine selected mechanical properties during a static tensile test (considering the results obtained using Equations (6) and (7)). The static tensile test also makes it possible to determine the relative reduction of area at fracture Z, which is also an important parameter when assessing whether the steel can be processed by cold plastic processing.

## 4. Directions of Further Research

In the future, we have planned to carry out the physical modelling of the analyzed process with the use of a complex deformation scheme (simultaneous tension and torsion) during the modelling of the rolling stages in NTM and RSM blocks of a rolling mill. The purpose of these tests is to create a strong deformation (neck) location in the deformed samples. Based on the results of many experimental and theoretical studies, it has been proven that the deformation speed in the neck during a tensile test is higher than the average speed calculated on the basis of changes in the measurement length of samples [41]. Taking advantage of this phenomenon will result in the obtained deformation speed being at a level at which, according to the literature data [32], the plasticizing stress of the tested material will not show significant changes.

An increase in the deformation speed in the analyzed process should positively affect the stress and activation of microstructure reconstruction processes and consequently increase the accuracy of the results of physical modelling (ferrite grain size and mechanical properties) in relation to the results of industrial tests. However, this will require designing a new geometry of samples, as the current geometry leads to a deformation location outside the research area, which disqualifies it for further analysis (Figure 18). The location of deformation outside the research area is also caused, in the analyzed case, by the dynamically changing temperature at this stage of the analyzed rolling process (rapid overheating after accelerated cooling between the NTM and RSM blocks of a rolling mill and the related inertia of the heat conduction phenomenon) and the geometry of the induction coil used in the STD 812 torsion plastometer.

In addition, we have planned to carry out tests of the analyzed process with alternating torsion and alternating compression and tensile tests.

## 5. Summary and Conclusions

The speed of implementing the results of theoretical calculations and laboratory scale tests in industry determines the development and dissemination of new technologies. Industrial research constitutes the final and very costly part of the implementation process, costly due to involving significant levels of production, amounts of manpower and materials, and the consumption of utilities. The costs of implementing new technologies can be significantly reduced and the process itself can be simplified and accelerated with the use of modern research methods. Physical modelling of the dynamic wire rod rolling in modern rolling mills is a complicated issue. This is due to, among others, high deformation speeds in individual rolling stands and short intervals between successive deformations. Therefore, during physical modelling of this process, it is necessary to take advantage of certain simplifications, for example, by replacing the sequence of several deformations with a single deformation. In the case of the analyzed, dynamic plastic treatment processes, such a solution is acceptable, as evidenced by the research results published, among others, in works [17,40].

Results of the parameters of the deformation process from physical modelling that are similar to those occurring in actual processes ensures the high accuracy of the obtained results, and they affect the shape and nature of changes in the plasticizing stress of the tested material and consequently the microstructure and properties of the finished product. In this paper, a similar value of plasticizing stress to that occurring in the actual rolling process [17] was achieved by changing the deformation pattern during the final stage of physical modelling from non-free torsion to the tension (variant V3). This is crucial for activating the microstructure reconstruction processes during the controlled cooling process immediately after the deformation is completed.

On the basis of the results presented in the paper, the following conclusions can be drawn:The best variant is the V3 variant, which consists of physically replicating the rolling process in a bar rolling mill as multi-sequence non-free torsion; the rolling process in an NTM block (No-Twist Mill) as non-free continuous torsion, with the total deformation equal to the actual deformation occurring during this stage of the technological process; and the rolling process in an RSM block (reducing and sizing mill) as a tension, while maintaining the total deformation value in this block;The difference between the ferrite grain size measured in the cross-section of a wire rod and obtained as a result of physical modelling according to the V3 technological variant was 4.3%, whereas the error between the ferrite grain sizes measured in the longitudinal cross-section of a wire rod and obtained as a result of physical modelling for this technological variant was 11.2%. On this basis, it can be concluded that the average ferrite grain size in a wire rod obtained under industrial conditions is similar to the grain size obtained in samples after physical modelling of the rolling process according to variant V3;The differences between the most important mechanical parameters determined during a static tensile test of a wire rod under industrial conditions and material after physical modelling were 1.5% for the yield strength, approximately 6.1% for the tensile strength, and approximately 4.1% for the constriction and plasticity reserve;The developed research methodology (variant V3) allows for replicating the entire deformation cycle while precisely preserving the temperature value of the deformed material during individual stages of the analyzed technological process and allows for quickly and accurately determining the most important mechanical properties during a static tensile test;Using a variable deformation scheme increases the research possibilities of modern torsion plastometers in terms of physical modelling of dynamic thermal and plastic treatment processes.

## Figures and Tables

**Figure 1 materials-16-00578-f001:**
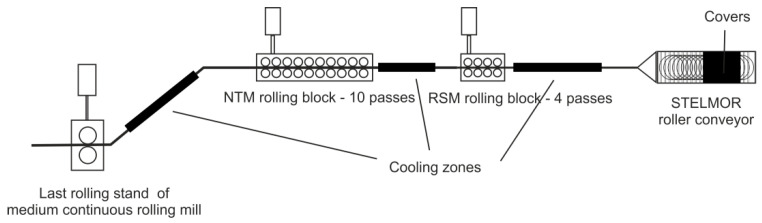
General scheme of the analyzed combined-type rolling mill [17]. In order to obtain a finished product with an even fine-grained ferritic–perlithic microstructure without a clear band structure, the final stage of deformation should take place in the austenitic range, when its temperature is 30–80 °C higher than the initial temperature of the austenite transformation, Ar_3_ [31,32,33,34,35]. For 20MnB4 steel, the Ar_3_ temperature is 780 °C. In the case of low-carbon and low-alloy steels, which are intended for further cold plastic processing, the most advantageous temperature for forming coils is a temperature of about 850–900 °C. Such a method of laying the coils provides an increased plasticity of the metal, beneficial for the cold drawing process, and allows a decrease in the recrystallizing annealing time after the drawing process [32].

**Figure 2 materials-16-00578-f002:**
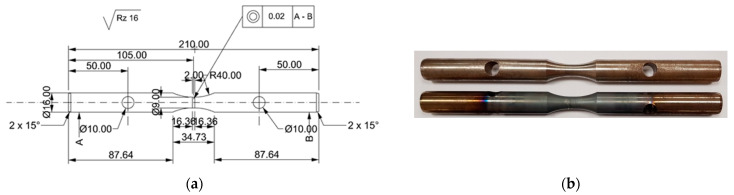
Samples for physical modelling tests: (**a**) technical specification and (**b**) general view of 20MnB4 steel samples before and after physical modelling.

**Figure 3 materials-16-00578-f003:**
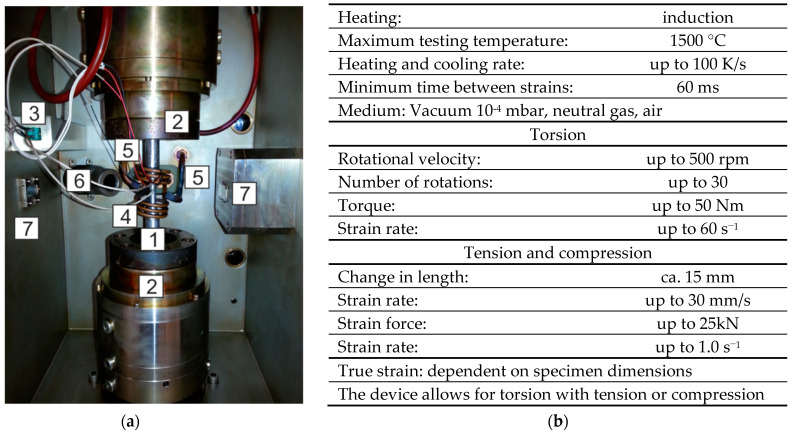
STD 812 torsion plastometer: (**a**) device chamber: 1—specimen, 2—holders, 3—thermocouples type S, 4—induction solenoid, 5—cooling system jets, 6—pyrometer, 7—sensors for laser measurement of specimen diameter and (**b**) basic specification [2,27].

**Figure 4 materials-16-00578-f004:**
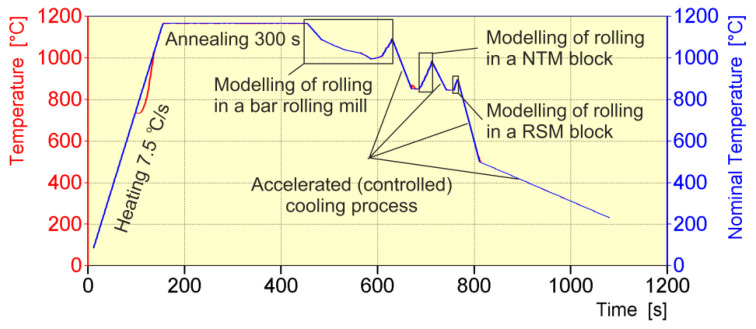
General model of thermo-mechanical treatment representing the entire rolling process of a 20MnB4 steel wire rod with a diameter of 5.5 mm.

**Figure 5 materials-16-00578-f005:**
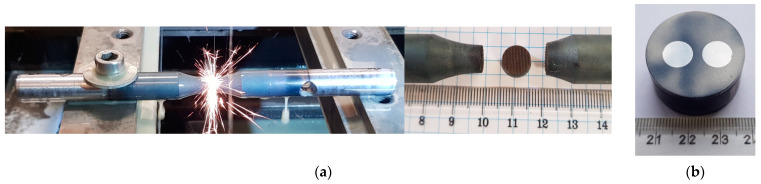
Method of sampling for metallographic tests from material after physical modelling: (**a**) general view and (**b**) sample metallographic microsections.

**Figure 6 materials-16-00578-f006:**
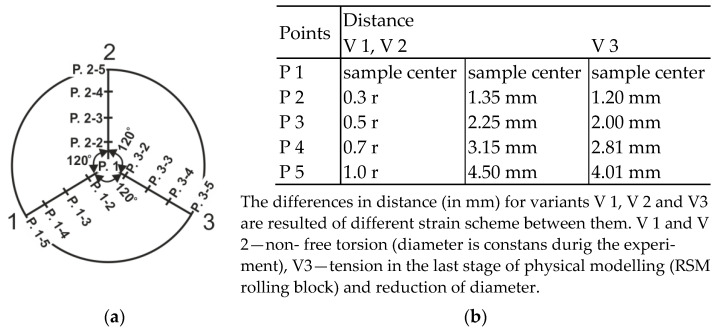
Cross-section of the material after physical modelling of the wire rod rolling process including marked measuring points: (**a**) general view and (**b**) distance table.

**Figure 7 materials-16-00578-f007:**
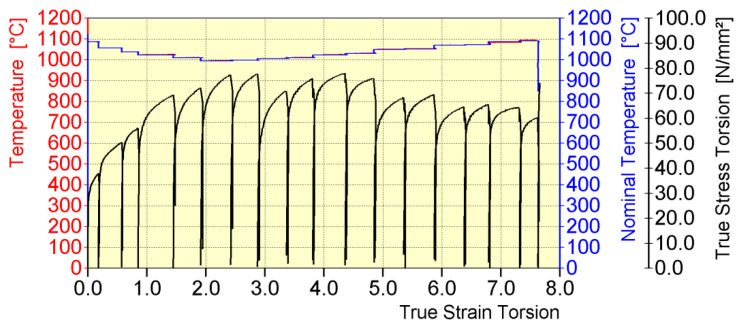
General course of temperature (nominal and obtained) and yield stress during physical modelling of rolling round 20MnB4 steel bars in a medium continuous rolling mill, for rolling a 5.5 mm diameter wire rod (test variants: V1, V2, and V3).

**Figure 8 materials-16-00578-f008:**
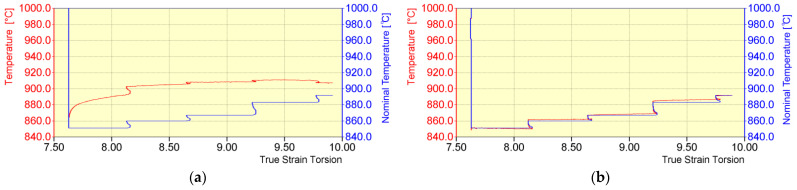
Example temperature distribution (nominal and obtained) during the initial stage of the deformation process in an NTM block: (**a**) accelerated cooling time before the NTM block (8.5 s) and (**b**) accelerated cooling time before the NTM block (55 s: accelerated cooling for 40 s and holding for 15 s).

**Figure 9 materials-16-00578-f009:**
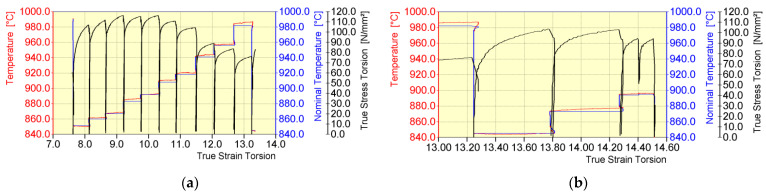
General course of temperature (nominal and obtained) and yield stress during physical modelling of rolling a 20MnB4 steel wire rod with a diameter of 5.5 mm (test variant V1): (**a**) in an NTM block and (**b**) in an RSM block.

**Figure 10 materials-16-00578-f010:**
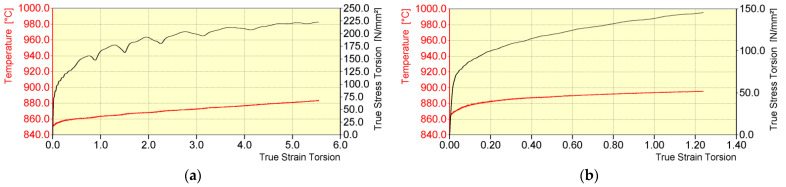
General course of temperature (nominal and obtained) and yield stress during physical modelling of rolling a 20MnB4 steel wire rod with a diameter of 5.5 mm (test variant V2): (**a**) in an NTM block and (**b**) in an RSM block.

**Figure 11 materials-16-00578-f011:**
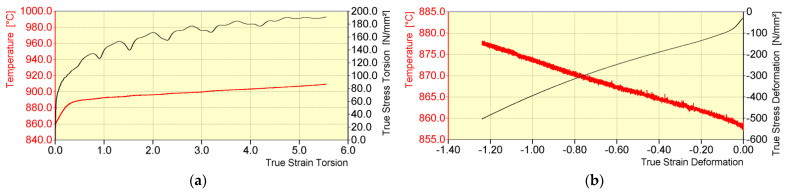
General course of temperature (nominal and obtained) and yield stress during physical modelling of rolling a 20MnB4 steel wire rod with a diameter of 5.5 mm (test variant V3): (**a**) in an NTM block (torsion) and (**b**) in the RSM block (tension).

**Figure 12 materials-16-00578-f012:**
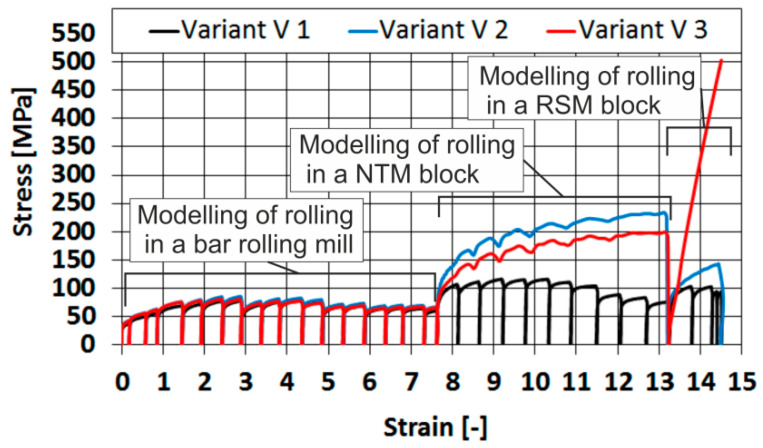
General course of yield stress during physical modelling of rolling a 20MnB4 steel wire rod with a diameter of 5.5 mm (test variant V1, test variant V2, test variant V3.

**Figure 13 materials-16-00578-f013:**
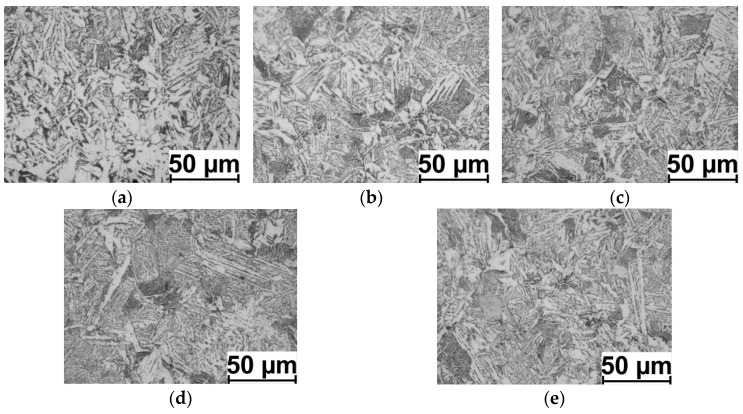
Example microstructures of 20MnB4 steel after physical modelling of a wire rod rolling process for test variant V1 (marking of points in accordance with Figure 5): (**a**) point P. 1, (**b**) point P. 1-2, (**c**) point P. 1-3, (**d**) point P. 1-4 and (**e**) point P. 1-5.

**Figure 14 materials-16-00578-f014:**
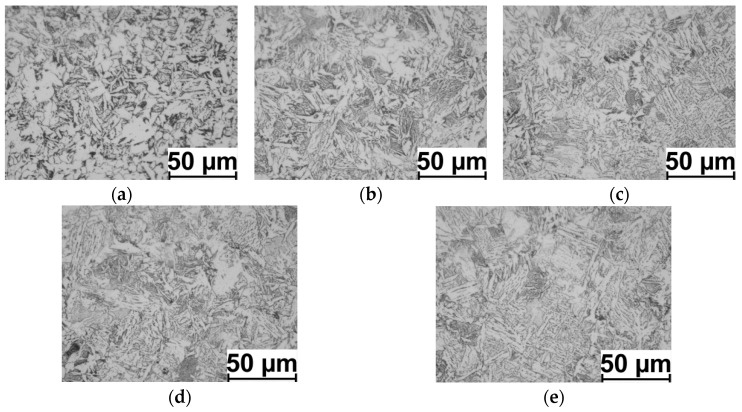
Example microstructures of 20MnB4 steel after physical modelling of a wire rod rolling process for test variant V2 (marking of points in accordance with Figure 5): (**a**) point P. 1, (**b**) point P. 1-2, (**c**) point P. 1-3, (**d**) point P. 1-4, and (**e**) point P. 1-5.

**Figure 15 materials-16-00578-f015:**
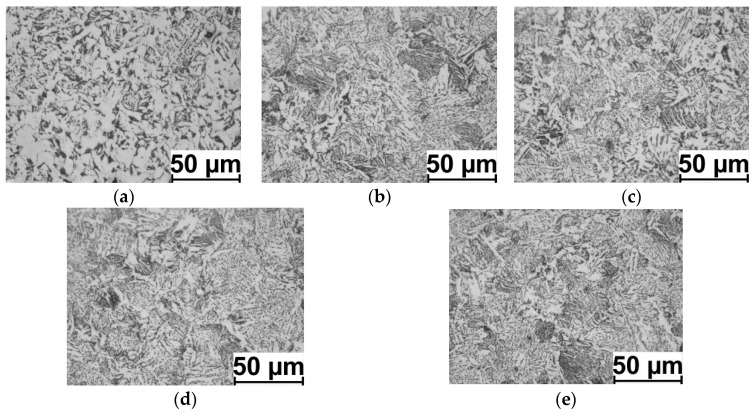
Example microstructures of 20MnB4 steel after physical modelling of the wire rod rolling process for test variant V3 (marking of points in accordance with Figure 5): (**a**) point P. 1, (**b**) point P. 1-2, (**c**) point P. 1-3, (**d**) point P. 1-4, and (**e**) point P. 1-5.

**Figure 16 materials-16-00578-f016:**
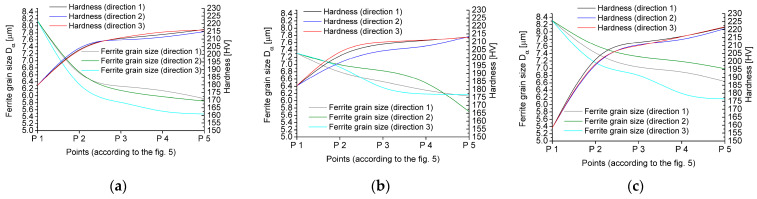
Distribution of the ferrite grain size and hardness of 20MnB4 steel after physical modelling of the rolling process of a wire rod with a diameter of 5.5 mm: (**a**) test variant V1, (**b**) test variant V2, and (**c**) test variant V3.

**Figure 17 materials-16-00578-f017:**
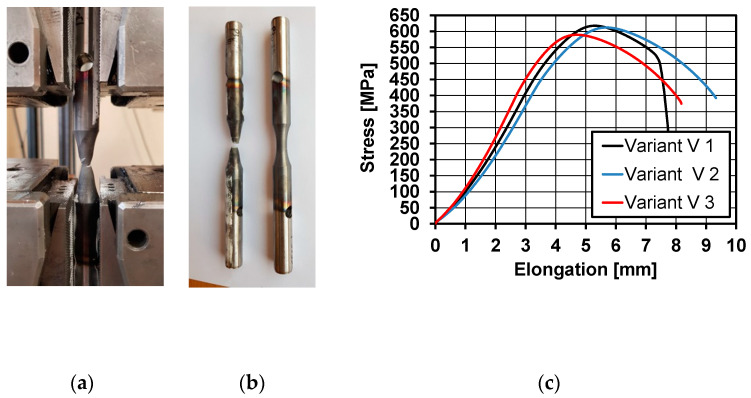
Sample results from testing the mechanical properties of the material after physical modelling of rolling a wire rod: (**a**) general view of a 20MnB4 steel sample after physical modelling, during determination of mechanical properties in a static tensile test, (**b**) general view of samples before and after the test, and (**c**) sample tensile curves of the tested material after physical modelling.

**Figure 18 materials-16-00578-f018:**
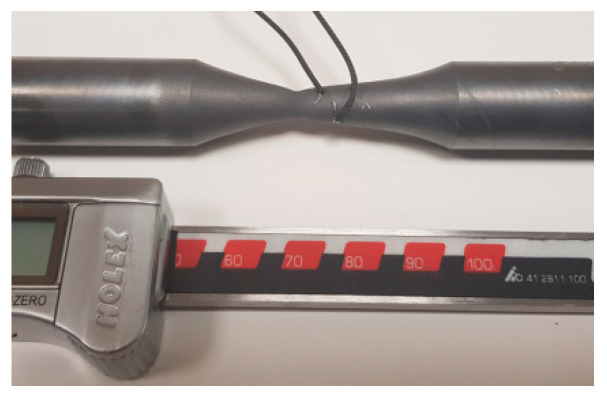
The view of the 20MnB4 steel sample after the deformation process in a complex deformation state (simultaneous torsion and tension) and location of the deformation outside the test area.

**Table 1 materials-16-00578-t001:** Chemical composition of 20MnB4 steel [17,21,30].

Steel Grade	Steel Number	Melt Analysis, Mass%						
20MnB4	1.5525	C	Si	Mn	P_max_, S_max_	Cr	Cu_max_	B
		0.18–0.23	≤0.30	0.90–1.20	0.025	≤0.30	0.25	0.0008–0.005

**Table 2 materials-16-00578-t002:** Parameters of the deformation process during physical modelling of rolling a round 20MnB4 steel rod in a medium continuous rolling mill, for rolling a 5.5 mm diameter wire rod (research variants: V1, V2, and V3) [17,21].

Pass Number	Temperature, *T* (°C)	$\mathrm{Strain},\;\varepsilon_{}\;(-)$	$\mathrm{Strain}\;\mathrm{Rate},\;{\dot{\varepsilon }}_{}\;(s^{\text{–}1})$	$\mathrm{Yield}\;\mathrm{Stress}\;(\mathrm{Torsion}),\;\sigma_{p}\;(\mathrm{MPa})$	Break Time After Deformation, *t* (s)
1	1086/1084	0.18	0.16	29.3	26.47
2	1057/1056	0.39	0.35	42.9	19.89
3	1037/1035	0.28	0.39	60.5	29.98
4	1023/1022	0.59	0.96	69.9	11.33
5	1010/1008	0.46	1.15	74.9	8.91
6	995/993	0.50	2.02/1.15	79.8	6.13
7	997/996	0.45	2.45/1.15	84.9	11.65
8	1005/1004	0.48	4.71/1.15	72.2	3.35
9	1009/1009	0.44	5.57/1.15	90.0	2.63
10	1022/1022	0.54	10.39/1.15	90.7	1.85
11	1030/1029	0.48	12.07/1.15	93.6	3.09
12	1049/1048	0.50	20.53/1.15	73.5	2.28
13	1052/1049	0.51	24.74/1.15	84.8	3.18
14	1069/1068	0.50	46.34/1.15	69.9	1.35
15	1072/1070	0.41	47.13/1.15	83.0	1.11
16	1087/1080	0.51	79.93/1.15	80.1	0.90
17	1091/1092	0.31	70.63/1.15	78.9	8.52/55.0

**Table 3 materials-16-00578-t003:** Parameters of the deformation process during physical modelling of rolling a 20MnB4 steel wire rod with a diameter of 5.5 mm in NTM and RSM blocks of a wire rod rolling mill (test variant V1) [17,21].

Pass Number	Temperature, *T* (°C)	$\mathrm{Strain},\;\varepsilon_{}\;(-)$	$\mathrm{Strain}\;\mathrm{Rate},\;{\dot{\varepsilon }}_{}\;(s^{\text{–}1})$	$\mathrm{Yield}\;\mathrm{Stress}\;(\mathrm{Torsion}),\;\sigma_{p}\;(\mathrm{MPa})$	Break Time after Deformation, *t* (s)
			NTM		
18	851/849	0.49	156.02/1.15	106.4	0.091/2.0
19	860/860	0.51	171.25/1.15	141.8	0.074/2.0
20	867/866	0.56	276.33/1.15	147.7	0.058/2.0
21	883/883	0.54	303.93/1.15	143.3	0.048/2.0
22	892/890	0.56	477.46/1.15	145.3	0.037/2.0
23	908/908	0.53	584.28/1.15	139.1	0.032/2.0
24	918/918	0.62	991.51/1.15	131.4	0.024/3.0
25	941/942	0.57	1042.10/1.15	106.3	0.020/3.0
26	956/955	0.62	1753.46/1.15	97.40	0.015/3.0
27	982/983	0.56	1809.67/1.15	86.5	0.82/45.0
RSM					
28	845/842	0.53	2368.05/1.15	96.0	0.012/3.0
29	873/874	0.48	2275.43/1.15	126.9	0.007/3.0
30	894/893	0.13	1853.11/1.15	111.9	0.004/0.9
31	895/894	0.10	1680.68/1.15	42.0	

**Table 4 materials-16-00578-t004:** Parameters of the deformation process during physical modelling of rolling a 20MnB4 steel wire rod with a diameter of 5.5 mm in NTM and RSM blocks of a wire rod rolling mill (test variant V2) [17,21].

Pass Number	Temperature, *T* (°C)	$\mathrm{Strain},\;\varepsilon_{}\;(-)$	$\mathrm{Strain}\;\mathrm{Rate},\;{\dot{\varepsilon }}_{}\;(s^{\text{–}1})$	$\mathrm{Yield}\;\mathrm{Stress}\;(\mathrm{Torsion}),\;\sigma_{p}\;(\mathrm{MPa})$	Break Time after Deformation, *t* (s)
NTM					
18	851–982/851–883	5.56	156.02–1809.67/40	222.8	0.82/6.0
RSM					
19	845–895/853–895	1.24	2368.05–1680.68/10	145.4	

**Table 5 materials-16-00578-t005:** Parameters of the deformation process during physical modelling of rolling a 20MnB4 steel wire rod with a diameter of 5.5 mm in NTM and RSM blocks of a wire rod rolling mill (test variant V3) [17,21].

Pass Number	Temperature, *T* (°C)	Strain, (-)	$\mathrm{Strain}\;\mathrm{Rate},\;{\dot{\varepsilon }}_{}\;(s^{\text{–}1})$	$\mathrm{Yield}\;\mathrm{Stress},\;\sigma_{p}\;(\mathrm{MPa})$	Break Time after Deformation, *t* (s)
NTM (torsion)					
18	851–982/856–909	5.56	156.02–1809.67/40	191.0	0.82/6.0
RSM (tension)					
19	845–895/858–878	1.24	2368.05–1680.68/10	503.0	

**Table 6 materials-16-00578-t006:** Results of measurements of ferrite grain size and hardness of 20MnB4 steel after physical modelling of the rolling process of a 5.5 mm diameter wire rod.

Points (According to Figure 6)	Variant V1		Variant V2		Variant V3	
	Ferrite Grain Size, D_α_ (μm)	Hardness (HV)	Ferrite Grain Size, D_α_ (μm)	Hardness (HV)	Ferrite Grain Size, D_α_ (μm)	Hardness (HV)
P.1	8.14	179.75	7.31	182.37	8.31	158.30
P. 1-2	6.38	206.83	6.73	203.90	7.31	209.20
P. 1-3	6.27	210.70	6.54	209.17	6.99	211.90
P. 1-4	6.17	212.78	6.27	210.77	6.94	215.30
P. 1-5	5.92	216.33	6.13	213.13	6.64	221.48
P.1	8.14	179.75	7.31	182.37	8.31	158.30
P. 2-2	6.46	208.83	6.95	198.43	7.55	204.20
P. 2-3	6.12	209.17	6.85	205.33	7.29	211.40
P. 2-4	5.97	210.97	6.60	206.40	7.21	212.83
P. 2-5	5.85	214.90	5.70	213.43	6.99	220,65
P.1	8.14	179.75	7.31	182.37	8.31	158.30
P. 3-2	6.00	206.03	7.00	207.23	6.93	205.50
P. 3-3	5.84	211.03	6.25	210.20	6.93	209.77
P. 3-4	5.51	215.13	6.18	210.87	6.19	215.50
P. 3-5	5.48	215.97	6.17	213.00	6.16	221.80
Average value	6.43	205.19	6.62	203.26	7.20	202.30

**Table 7 materials-16-00578-t007:** Measurement results of selected mechanical properties of 20MnB4 steel.

Research Variant of Physical Modelling	Mechanical Properties
Calculated Using Formulas:	Static Tensile Test
(4)	(5)	(4), (5)	(6)	(7)	(6), (7)
Yield Strength,YS (MPa)	Ultimate Tensile Strength, UTS (MPa)	YS/UTS	Yield Strength,YS (MPa)	Ultimate Tensile Strength, UTS (MPa)	YS/UTS	Yield Strength,YS (MPa)	Ultimate Tensile Strength, UTS (MPa)	YS/UTS	RelativeReduction of Area at Fracture, Z (%)
Variant V1	420	653	0.64	426	566	0.75	429	605	0.71	62.03
Variant V2	415	648	0.64	422	564	0.75	422	598	0.71	57.49
Variant V3	412	645	0.64	412	558	0.74	**418**	**591**	**0.71**	**66.93**
Physical modelling in compression test [17,21]	401	632	0.63	397	550	0.72				
Industrial research results		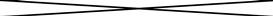		**412**	**557**	**0.74**	**69.80**

## Data Availability

The data presented in this study are available on request from the corresponding author.

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
