# Peer review of "Innovative Methodology for Physical Modelling of Multi-Pass Wire Rod Rolling with the Use of a Variable Strain Scheme"

_materials, 2023, doi:10.3390/ma16020578_

Round 1
Reviewer 1 Report
Laber K. has tried to present the results of the physical modelling of the process of multi-pass rolling of wire rod with controlled, multi-stage cooling in the manuscript entitled “Innovative methodology for physical modelling of multi-pass wire rod rolling with the use of a variable strain scheme” since the industrial research constitute the final but very costly part of the 697
implementation process. While the title mentioned Innovative methodology…”, I think it is not a novel innovative method many publications have been published on this topic and the author has been overclaimed. Moreover, while the manuscript has a similarity rate of 27% to the previously published article, more than 13% is related to the author's published article in the same journal “https://doi.org/10.3390/ma13030711”.
Aside from this, abstract and conclusion are well-stablished. However, I would like to draw the author's attention to the following issues:
1. Overall, there is lack of novelty is this study.
2. Table 1, is the same material used in the author’s previous article.
3. Line 153 “30÷80°C” what is the meaning of “÷”. It also has been used in other parts of the manuscript and tables.
4. Figure 2 has been borrowed from the author’s article entitled “The problems of physical modelling of the processes of wire rod rolling at high rolling velocities” with the Corpus ID: 55930492.
5. There is lack of compression between the finding of this study and previous literature.
Author Response
Response to Reviewer 1 Comments (Round 1)
Point 1: Overall, there is lack of novelty is this study.
Response 1: Thank you very much for this question. Looking through the technical literature, I did not find works in which a similar process of multi-pass rolling with controlled (accelerated) cooling, was modeled by using a torsion plastometer with the use of a variable strain scheme. As I wrote in the introduction, most of the works concerning the physical modelling of the rolling processes, published so far, focuses on the assumption that the microstructure and properties of the final product are mainly determined by the few final passes, and the impact of the earlier stages of the technological process is less important [15]. As shown by the results of the research presented, inter alia, in the works [16-21], such a simplification is acceptable. Therefore, physical modelling studies are usually carried out only for the final stage of the analyzed technological process, often during compression tests [16-21]. The biggest advantage of this research methodology consists in the possibility of obtaining large values of strain rate, which significantly affects the level of yield stress. On the other hand, the biggest disadvantage is the limited value of total strain value (about 1.2) and sometimes limitations in the scope of precisely controlling the process of accelerated (controlled) cooling, which is important in the case of modelling processes which include the so-called multi-stage inter-operational cooling. Moreover, the material after physical modelling often cannot be used to determine mechanical properties directly during a tensile test and determining mechanical properties takes place indirectly, based on measuring the grain size, hardness, and chemical composition [17, 20, 21], which is time-consuming and may include some error.
Personally, I have also used this methodology so far.
In this article, I wanted to check whether the applied methodology can be used for the analysed process and what results (accuracy) it will give. Hence the word “innovative” in the title.
In my opinion the advantage (novelty) of the research methodology proposed in the paper in reference to the works published so far is the possibility of reproducing the entire strain cycle (total strain value 14.32) with precisely preserving the temperature value of the deformed material during individual stages of the reproduced technological process and the possibility of quickly and accurately determining selected mechanical properties during a static tensile test, directly from the material after physical modelling – as I wrote in the introduction.
The analyzed problem will be further developed with the use of a complex deformation scheme (simultaneous tension and torsion) during the modelling of the rolling stages in NTM and RSM blocks of a rolling mill. Moreover, it is planned to carry out tests of the analyzed process in alternating torsion and alternating compression and tensile tests.
Moreover, in my opinion using a variable deformation scheme increases the research possibilities of modern torsion plastometers in terms of physical modelling of dynamic thermal and plastic treatment processes.
Point 2: Table 1, is the same material used in the author’s previous article.
Response 2: Thank You very much for comment. You have right. It is the same material.
In this work, I gave a reference to the standard to indicate that the chemical composition of the steel is in accordance with the standard. I am working with the same material to have as many results as possible to comparison of the obtained results (after different modification). I corrected this error at this work by adding appropriate (additional) references [17, 21]
Point 3: Line 153 “30÷80°C” what is the meaning of “÷”. It also has been used in other parts of the manuscript and tables.
Response 3: The symbol “÷” denotes a range: in this case from 30°C to 80°C higher temperature value than the initial temperature of the austenite transformation Аr3. This symbol has been changed to the “-“ in the whole article.
Point 4: Figure 2 has been borrowed from the author’s article entitled “The problems of physical modelling of the processes of wire rod rolling at high rolling velocities” with the Corpus ID: 55930492.
Response 4: Thank You very much for this comment. You have right. This is my own figure that was published in my previous paper [27]. I forgot that it was also published in paper no [2]. I added this reference also to this paper.
Point 5: There is lack of compression between the finding of this study and previous literature.
Response 5: I'm not sure if I understood the question correctly – do you mean that there is no comparison of the results presented in this paper with previous results? I want to explain that results that was obtained in this paper (especially variant V 3) were compared with previous results, presented, among other things in the paper [17, 21]. Obtained in this paper results were compared with industrial research results also. Comparison results are presented in the table no 7.
Additionally the manuscript was verified in grammar by professional translator company (www.dogadamycie.pl).

Reviewer 2 Report
The article is devoted to the physical modeling of a very complex technological process, namely, hot rolling of 5.5 mm wire rod. The rolling process is high-speed and the exit speed of the last rolling stand is more than 50 m/s. It is economically and technically difficult to develop and improve wire rod production modes on real equipment. At the same time, the rolling and cooling modes of wire rod determine the level of mechanical properties and the quality of the billet for subsequent processing. Therefore, the relevance of the work is not in doubt. Methods and equipment that are used in the research allow to obtain reliable results. The article is very well designed and structured. It is written very clearly and technically well. High quality graphic material. There is a scientific novelty in the work, which consists in studying the regularities of the influence of thermo-mechanical treatment of 20MnB4 steel on the formation of the microstructure and mechanical properties. The presented paper, in my opinion, is a finished article that can be published in the highly rated journal "Materials" without additional corrections, however, I have a few comments to the author.
1. In the Introduction or 2.2 Methods, after lines 145...150, it would be useful to give a simplified scheme of a rolling mill for the production of wire rod. This would be helpful to readers who are not very familiar with this production.
2. Researchers of hot deformation now very often use the simulator of thermomechanical processes Gleeble 3500 or 3800 for physical modeling. The author did not mention this method in the Introduction.
Author Response
Response to Reviewer 2 Comments (round 1)
Dear Reviewer
Thank You very much for the review and all comments. Answers for Your comments are bellow.
Point 1: In the Introduction or 2.2 Methods, after lines 145...150, it would be useful to give a simplified scheme of a rolling mill for the production of wire rod. This would be helpful to readers who are not very familiar with this production.
Response 1: Thank you very much for this comment. You have right, additional scheme of the analysed rolling line will be helpful to readers who are not very familiar with this production. I added this scheme as a figure no 1. The other figures have been renumbered accordingly.
Point 2: Researchers of hot deformation now very often use the simulator of thermomechanical processes Gleeble 3500 or 3800 for physical modeling. The author did not mention this method in the Introduction.
Response 2: Thank you very much for comment. I agree with this comment. I wrote about this methodology in the introduction – lines 55-74. But I didn’t wrote that these research were made by using Gleeble System. Now I added this short information to the introduction.

Reviewer 3 Report
In my view, paper is of interest. Here are some comments:
1. Page 3, Materials and Methods. Which rolling mill was chosen as an object for the investigation? To which plant does it belong? Where is it located?
2. Page 4, lines 144-149. You mention the final diameter of the rod. What was the initial diameter? I did not find it in the paper. Correct me if I am wrong.
3. The scheme of Figure 5a is a little bit hard to understand. All the three radii are in the same cross section, right? Angles between them are all equal to 120 degrees, is it correct? Distance between the points on every radius is not equal? Could you, please, explain, how location of the points on the radius for V1, V2, and V3 was chosen?
4. Page 17, line 567: “…relatively high homogeneity…” – could you, please, characterize it numerically?
5. Page 17, line 571: “…relatively large heterogeneity…” - could you, please, characterize it numerically? In this line you also mention the “acicular shape” of the grains. Does it suppose anisotropy of the grain size? Did you estimate it? If yes, please, provide the numerical results.
6. Page 19, Table 7. The last line, “Industrial research results” – the cell of this line that corresponds to “Calculated by using formulas” column is empty. It means that there is no data for that cell? Or you forgot to fill it? Please, specify.
7. Page 20, line 651 – “lowest accuracy”; and page 20, line 655 – “higher accuracy” – numerical values proving mentioned extents of accuracy are desirable.
Author Response
Response to Reviewer 3 Comments (Round 1)
Dear Reviewer. Thank You very much for Your deep review and all questions and comments. My answers are bellow.
Point 1: Page 3, Materials and Methods. Which rolling mill was chosen as an object for the investigation? To which plant does it belong? Where is it located?
Response 1: I can’t answer which rolling mill department exactly it is . I can’t also write this information in the paper. But I can explain that presented in the paper results are for combined type rolling mill (bar rolling mill and Morgan wire-rod rolling mill). I added short scheme on figure no. 1. This rolling line is located in Poland.
Point 2: Page 4, lines 144-149. You mention the final diameter of the rod. What was the initial diameter? I did not find it in the paper. Correct me if I am wrong.
Response 2: I can’t to add this information in my paper. The Steelworks dosen’t agree for publication these detailed information. But only for Your information I can write that the wire rod was rolled from a stock from continuous casting process – square with cross section 160 mm. After pass No. 17 (input dimension to NTM rolling block), the round rod had a 20 mm in diameter.
Point 3: The scheme of Figure 5a is a little bit hard to understand. All the three radii are in the same cross section, right? Angles between them are all equal to 120 degrees, is it correct? Distance between the points on every radius is not equal? Could you, please, explain, how location of the points on the radius for V1, V2, and V3 was chosen?
Response 3: Yes You have right. All the three radii are in the same cross section (center of the sample length (working area)). Angles between them are 120 degrees. I added angle value on the figure 5a and short explanation below the figure. I hope that now it is more understandable for readers. I chosen this values to determine the distribution of hardness and ferrite grain size across the radius. In each direction, measurements were made at 5 points. To increase the accuracy – I choosed three radius (in three directions). The differences in distance (in mm) for variants V 1, V 2 and V3 are resulted of differend strain scheme between them. In Variants V 1 and V 2 there was non- free torsion (diameter is constans durig the experiment) but for variant V3 there was tension in the last stage of physical modelling (RSM rolling block) and decrease of diameter. This is the reason why there is differences between the value (in mm) for analysed points.
Point 4 and 5: Page 17, line 567: “…relatively high homogeneity…” – could you, please, characterize it numerically? Page 17, line 571: “…relatively large heterogeneity…” - could you, please, characterize it numerically? In this line you also mention the “acicular shape” of the grains. Does it suppose anisotropy of the grain size? Did you estimate it? If yes, please, provide the numerical results.
Response 4 and 5: Indeed, the terms “relatively high homogeneity” or “relatively large heterogeneity” are imprecise. I meant that taking into account the characteristics of the torsion test itself, which is characterized by a large unevenness of deformation parameters on the cross-section, I expected greater differences in grain size between points p.1-p.5. However, the differences are smaller than I thought. As I wrote, this results from many factors: the size of the total deformation, the dimensions of the working part of the sample and the dynamics of temperature changes over the time. I do not currently have any numerical results to prove this. I made my conclusions based on metallographic tests (only light microscopy) and hardness tests. However, when it comes to the grain size at individual points, as I wrote, it is not homogeneous. This is also due to several things. Among other things, the dynamics of the microstructure remodelling processes in individual areas and the cooling rate after the entire process. 10 C/s is the limit speed of cooling for the tested steel. As my previous research [17] showed, after exceeding the cooling rate of 10/s, bainite structures are formed in this steel grade. In the future, I plan to investigate these phenomena numerically and perhaps with the use of electron microscopy. I want to investigate the effect of stress on microstructure remodelling processes. Numerical mapping of the entire process, taking into account all parameters of thermoplastic processing, is very complicated. Much depends on the numerical models of the rheological properties of the material itself, which was shown for another material in [27]. The numerical program that I intend to use allows for the analysis of deformation parameters. It models issues related to microstructure changes in a worse way. At the moment I have not studied the anisotropy of the grain.
Point 6: Page 19, Table 7. The last line, “Industrial research results” – the cell of this line that corresponds to “Calculated by using formulas” column is empty. It means that there is no data for that cell? Or you forgot to fill it? Please, specify.
Response 6: Tests of the mechanical properties of the wire rod in industrial conditions were carried out on the basis of a static tensile test only. The properties of the real wire rod were not tested using analytical formulas. I crossed out these fields in the table no 7 to make it easier to understand.
Point 7: Page 20, line 651 – “lowest accuracy”; and page 20, line 655 – “higher accuracy” – numerical values proving mentioned extents of accuracy are desirable.
Response 7: I agree with this comment. Appropriate corrections have been made in this regard in the article.
